# Discovery and Characterization of Two Selective Inhibitors for a Mu-Class Glutathione S-Transferase of 25 kDa from *Taenia solium* Using Computational and Bioinformatics Tools

**DOI:** 10.3390/biom15010007

**Published:** 2024-12-25

**Authors:** César Sánchez-Juárez, Roberto Flores-López, Lluvia de Carolina Sánchez-Pérez, Ponciano García-Gutiérrez, Lucía Jiménez, Abraham Landa, Rafael A. Zubillaga

**Affiliations:** 1Departmento de Química, Universidad Autónoma Metropolitana-Iztapalapa, Mexico City C.P. 09310, Mexico; csj3140@xanum.uam.mx (C.S.-J.); lluvia_sanper@xanum.uam.mx (L.d.C.S.-P.); 2Departamento de Microbiología y Parasitología, Facultad de Medicina, Universidad Nacional Autónoma de México, Mexico City C.P. 04510, Mexico; robertof@comunidad.unam.mx (R.F.-L.); lucia.jimenez@facmed.unam.mx (L.J.); landap@servidor.unam.mx (A.L.); 3Posgrado en Ciencias Biológicas, Unidad de Posgrado, Universidad Nacional Autónoma de México, Mexico City C.P. 04510, Mexico

**Keywords:** glutathione S-transferase, selective inhibitors, molecular modeling

## Abstract

Glutathione S-transferases (GSTs) are promising pharmacological targets for developing antiparasitic agents against helminths, as they play a key role in detoxifying cytotoxic xenobiotics and managing oxidative stress. Inhibiting GST activity can compromise parasite viability. This study reports the successful identification of two selective inhibitors for the mu-class glutathione S-transferase of 25 kDa (Ts25GST) from *Taenia solium*, named *i11* and *i15*, using a computationally guided approach. The workflow involved modeling and refining the 3D structure from the sequence using the AlphaFold algorithm and all-atom molecular dynamics simulations with an explicit solvent. Representative structures from these simulations and a putative binding site with low conservation relative to human GSTs, identified via the SILCS methodology, were employed for virtual screening through ensemble docking against a commercial compound library. The two compounds were found to reduce the enzyme’s activity by 50–70% under assay conditions, while showing a reduction of only 30–35% for human mu-class GSTM1, demonstrating selectivity for Ts25GST. Notable, *i11* displayed competitive inhibition with CDNB, while *i15* exhibited a non-competitive inhibition type.

## 1. Introduction

Drug design is a multidisciplinary field that merges knowledge from chemistry, biology, bioinformatics, and pharmacology to develop more effective and specific therapeutic agents. While traditional drug discovery relied heavily on the empirical screening of large compound libraries to find candidates with therapeutic potential, modern approaches have shifted towards precision and specificity. Techniques such as structure-based and ligand-based design now enable the identification and optimization of compounds that interact with well-defined biological targets. This evolution has markedly reduced the time and cost of drug development, making the process more efficient and streamlined [1].

A central technique in contemporary drug design is molecular modeling, which utilizes computational tools to simulate and predict the interactions of drugs with biological targets. These simulations allow for the identification and optimization of promising candidate molecules before experimental validation, thus enhancing the efficiency of the design process [2]. Furthermore, advancements in artificial intelligence (AI) and machine learning (ML) are accelerating the identification of novel molecular structures with therapeutic potential and predicting their pharmacokinetic and pharmacodynamic properties [3]. This convergence of technologies has markedly increased productivity in the pharmaceutical sector [4].

Numerous studies underscore the success of computational strategies in identifying molecules with specific desired effects [5]. Such is the case with the inhibition of enzymes crucial for the survival of helminth parasites such as *Taenia solium*, in which glutathione S-transferases (GSTs) constitute their main detoxification system, catalyzing conjugation reactions between glutathione and a variety of endo- and exo-electrophilic substrates, which increase its solubility for subsequent excretion. To date, four GSTs have been identified in the cytoplasm of *T. solium*, the Ts26GST variant being the most abundant. All of them present the same thioredoxin-like fold with two structural domains. The N-terminal domain contains the site where the substrate GSH binds, called the G site (shown in Figure 1) and the C-terminal domain contains the binding site for electrophilic substrates, commonly hydrophobic, which is known as the H site. This is located next to the G site, allowing the formation of the conjugated molecule.

A study by García-Gutiérrez et al. (2020) identified a non-competitive inhibitor, termed *i7*, through virtual screening, targeting the 26 kDa glutathione S-transferase (Ts26GST) of *Taenia solium* [6]. This inhibitor interacts non-competitively with glutathione (GSH) and exhibits a mixed competitive mechanism with the universal electrophilic substrate 1-chloro-2,4-dinitrobenzene (CDNB), making it a promising candidate for antiparasitic drug development [6,7].

Ts26GST, an alpha/mu-class enzyme, is the major detoxifying enzyme of the four cytosolic GSTs in *T. solium*. The three additional GSTs are a 25.5 kDa mu class GST (Ts25GST), a 24.5 kDa sigma class GST, and a 27 kDa omega class GST (Ts27GST). All of them must be considered in the design of effective treatments [8,9,10]. While each GST variant plays a critical role in parasite detoxification systems [11,12], they may contribute to other essential functions, including the modulation of the host immune response (Ts24GST), management of oxidative stress (Ts25GST), or signaling (Ts27GST) [12,13,14,15,16]. This study presents the identification of two new selective inhibitors for the 25.5 kDa mu-class GST from *T. solium* with a low inhibitory activity towards its human homolog, the mu-class GST isoform M1 (HGSTM1). These inhibitors were identified and characterized through a combination of computational techniques, including homology modeling, molecular docking, and molecular dynamics, which facilitated the screening of 50,000 compounds from Chembridge’s DiverSet Express Pick library. The identified compounds offer a promising foundation for the development of a novel antiparasitic agent aimed at inhibiting Ts25GST, potentially advancing treatment strategies against *T. solium*.

## 2. Materials and Methods

### 2.1. Homology Modeling

A three-dimensional (3D) dimeric model of the 25.5 kDa mu-class glutathione S-transferase (Ts25GST) from *T. solium* was constructed using the UniProt sequence Q8MWS0. A homologous structure of the mu-class GST from *Gallus gallus* (PDB-ID 1GSU) [17], sharing 46.33% of the sequence identity and with a resolution of 1.94 Å, was identified in the Protein Data Bank. Using AlphaFold2 [18,19], five independent models were generated, from which the most reliable model was selected. Glutathione (GSH) was incorporated into the model through its alignment with homologous mu-class GST structures complexed with GSH (PDB-IDs 1XW5, 1XW6, and 2FHE), which exhibit sequence identities between 40% and 45%. The final model incorporates GSH within the binding site, constructed using the 1XW5.PDB file which has the best resolution (1.8 Å).

### 2.2. All-Atom Molecular Dynamics Simulations

Molecular dynamics (MD) simulations were conducted to explore the behavior of the Ts25GST-GSH complex in the solution. The simulations utilized the Amber99sb* force field, TIP4P-Ewald water model, and protonation states adjusted to pH 7.4 using PropKa. The atomic charges for the ligands, including GSH, were calculated using Acpype and Gaussian. The system was simulated in triplicate, each with different initial seeds, under NVT conditions at 310.15 K for 1.0 nanoseconds using the V-rescale thermostat. Pressure equilibration was achieved at 1 bar using the Parrinello–Rahman barostat for an additional 1.0 nanosecond. Production runs of 1.0 μs were performed and analyzed using GROMACS and TTclust in Python [20].

### 2.3. SILCS-Based Site Identification

To select the region for virtual screening, the SILCS (Site-Identification by Ligand Competitive Saturation) method [21,22,23,24] was employed. This involved three all-atom MD simulations of Ts25GST bound to GSH in co-solvent, using the OPLS-A force field with isopropanol as a probe (~0.2 M). Charges were adjusted for the TIP4P-ε water model. The simulations were conducted at 310.15 K for 100 ns with a 2.0 fs timestep. Three additional reference simulations in co-solvent without the protein were also performed. Results were analyzed using the volmap and voltool modules in VMD 1.94 [24].

### 2.4. Virtual Screening

An ensemble docking approach [25,26,27] was performed using structures generated by TTclust. Docking simulations were carried out with VinaMPI [28,29,30] and GOLD 2022.2 [31,32]. Protein and ligand preparations were performed using MOE v2014 [33], AutoDock Tools [34], and OpenBabel [35]. For ligand selection, energy cutoffs of −9.0 kcal/mol in Vina and Chem PLP scores above 75 were applied, based on re-docking studies of mu-class GSTs co-crystallized with inhibitors. The Diver-Set EXPRESS-PICK library from ChemBridge, containing 50,000 compounds, was screened in three stages as follows: low-exhaustiveness (Vina = 8, 10 GA runs), exhaustive screening (Vina = 32, 100 GA runs), and final docking against human mu-class GST (HGSTM; 1GTU, 1XW5, 1XW6, 3GTU, 4GTU) to identify selective inhibitors.

### 2.5. Expression and Purification of Recombinant Ts25GST

Ts25GST production was carried out following the protocol of Roldán et al. [9]. *E. coli* JM105 cells were transformed with the pTRc99A plasmid containing the Ts25GST gene. A pre-culture was grown in LB medium containing 100 μg/mL ampicillin at 37 °C. The culture was scaled up to 500 mL, and growth was monitored by measuring optical density at 600 nm (OD600) until it reached ~0.6. Overexpression of Ts25GST was induced with 2.0 mM IPTG and incubated for an additional 5 h. Protein was purified by affinity chromatography on a GSH-Sepharose 4B column according to the methods described by Torres-Rivera et al. (2008) and Vivanco-Pérez et al. (2002) [7,9]. The protein concentration was determined by the Bradford method, and the purity was confirmed by SDS-PAGE.

### 2.6. Enzyme Kinetics

Glutathione S-transferase activity was assessed in 100 mM Tris-HCl buffer, pH 7.4, with 20% DMSO, using a modified version of the Habig method [6,7,8,9,10,11,12]. Bi-substrate kinetics were evaluated by conducting two series of experiments as follows: in the first series, GSH was kept at 5.0 mM while CDNB concentrations ranged from 0 to 12 mM; in the second series, CDNB was held constant at 7.0 mM while GSH varied from 0 to 9.0 mM. Reactions were incubated at 37 °C for 30 min, and absorbance at 340 nm was monitored using the double-beam spectrophotometer Shimadzu UV-1800. A reaction without protein served as the blank. Initial reaction rates were calculated from the first 20 s of the kinetic curve using QtiPlot and fitted to the Hill model (Equation (2) for CDNB) [36,37] and the Michaelis–Menten model (Equation (3) for GSH) [38].

Equation (1), *V*_0_ calculation
(1)$$V_{0}=\frac{m\times (60)}{9.6\;\times 10^{-3\;}\times mg\;enzyme}$$
where

*V*_0_: initial velocity [mmol⋅min^−1^⋅mg^−1^];*m*: slope of the reaction progress curve during the first 30 s [AU s^−1^].

Equation (2), Michaelis–Menten model
(2)$$V_{0}=V_{max}\frac{S}{S+K_{M}}$$
where

*V_max_:* maximum velocity [mmol⋅min^−1^⋅mg^−1^];*K_M_:* Michaelis–Menten constant [mM];*S*: substrate concentration [mM].

Equation (3), Hill model
(3)$$V_{0}=V_{max}\frac{S^{n}}{S^{n}+{K_{M}}^{n}}$$
where

*n*: Hill coefficient.

### 2.7. In Vitro Activity Assays

In vitro inhibition assays were performed according to the protocol presented by García et al. [6], with minor modifications. Reactions were conducted in 100 mM Tris-HCl buffer, pH 7.4, containing 5.0 mM GSH and 20% DMSO. Recombinant Ts25GST at a concentration of 1.0 μg/mL was incubated at 37 °C for 30 min with 40 μM inhibitor, and the reaction was initiated by adding 8.0 mM CDNB. The reaction was monitored at 340 nm, and the specific activity was calculated within the first 20 s. Inhibition and residual activity were determined relative to the controls. Selectivity tests were performed using human (*Homo sapiens*) mu class M1 GST from Oxford Biomedical Research (GS65) under the same conditions as those for Ts25GST.

Equation (4), residual activity
(4)$$\%\;Residual\;activity=\frac{V_{0}\left(+inhibitor\right)}{V_{0}\left(-inhibitor\right)}*100$$

Equation (5), inhibition
(5)% inhibition=100−% Residual activity

### 2.8. Characterization of Inhibitory Effects

IC_50_ values for the selected inhibitors were determined using the conditions of the in vitro assays, varying the inhibitor concentration from 10 to 70 μM. A dose–response curve was generated by plotting the logarithm of the inhibitor concentration against the inhibition percentage, and the data were fitted to Equation (6) to calculate the IC_50_. Enzyme kinetics were then performed with GSH fixed at 5.0 mM and varying CDNB concentrations from 0 to 12 mM, in the presence of inhibitors from 10 to 30 μM. The Hill equation was used to fit the data, and kinetic parameters were compared between the control and inhibitor-treated conditions.

Equation (6), dose–response
(6)$$y=A1+\frac{A2-A1}{1+10^{\left(log(x_{0}-x\right)p}}$$
where

*y*: percentage of inhibition;*A*1: minimum value;*A*2: maximum value;*x*_0_: inflection point (IC50);*p*: slope of the curve;*x*: logarithm of the inhibitor concentration (Log [inhibitor]).

## 3. Results

### 3.1. Structural Modeling of Ts25GST

#### 3.1.1. Homology Modeling

Due to the lack of experimentally resolved 3D structures for Ts25GST, a structural model was generated using AlphaFold. The resulting model (Figure 1A) exhibits the typical thioredoxin-like fold of GSTs [13,14], with an N-terminal region containing the GSH-binding site and a C-terminal region comprising five alpha helices and loops where the xenobiotic binding site is located [9,10,11,12,13,14].

Analysis of the GSH-binding site revealed a high degree of conservation; of the 15 residues in the G site, only 4 showed variability in other mu-class GSTs, consistent with the known structural features of this enzyme family [13,14]. The structural superposition of Ts25GST with homologous mu-class GSTs complexed with GSH (Figure 1C) allowed us to position this substrate in our model. The 2D interaction diagrams are shown in the Appendix A (Appendix A), highlighting the key interactions. Tyrosine 7 aligns with tyrosines from homologous GSTs known to activate GSH, suggesting a similar role is assumed by Tyr7 in Ts25GST [9,10,14].

The model was validated via SAVES 6.1 (https://saves.mbi.ucla.edu/, URL accessed on 1 June 2022) with favorable metrics—ERRAT (93.4%) [39], Verify3D (91.2%) [40], and PROVE (2.3%) [41]—and no residues in the disallowed regions on the Ramachandran plot [42]. Additionally, a conserved mu loop was identified between alpha helix 2 and beta sheet 2 in the N-terminal domain [14].

Structural alignments with the unbound, GSH-bound, and HGSTM1-conjugated forms (Figure 1C) suggest a more open catalytic site for Ts25GST.

#### 3.1.2. All-Atom Molecular Dynamics Simulations of the Model

MD simulations showed consistent behaviors across trajectories; the Root Mean Square Deviations (RMSD) values ranged from 0.18 to 0.25 nm relative to the initial structure (Appendix A), the radii of gyration from 2.16 to 2.22 nm (Appendix A), and with similar Root Mean Square Fluctuations (RMSF) profiles (Appendix A). Principal Component Analysis (PCA) [43,44,45,46] indicated non-linearity in eigenvector 5 across replicates (Figure 2A), with the first five components explaining 54.2%, 53.4%, and 51.1% of the variance in replicates 1, 2, and 3, respectively. The temporal projection of eigenvalues (Figure 2B) revealed a dynamic convergence across trajectories.

Free energy landscape analysis using concatenated simulations of the first two PCA components (Figure 2C) identified a global minimum and two local minima. Trajectories within these minima were extracted and clustered yielding five clusters (Figure 2D) with intergroup RMSDs of 1.8–2.1 Å. The superposition of representative structures (Figure 2E) revealed significant variability in the mu loop, consistent with its known role in GST classification and catalytic function. The deletion of this loop in previous studies reduced substrate affinity without affecting structural stability or reaction rate in mammalian GSTs [14].

Analysis of GSH binding in representative structures (Figure 2F,G) indicated a stable interaction between GSH and Tyr7 in chain A, consistent with previous studies [6,7,8,9,10,11,12]. In this chain A, GSH adopted an extended conformation, like that observed in co-crystallized GST structures. In contrast, chain B displayed atypical GSH conformations in some trajectories. In 60% of the simulations, the distance between the sulfur atom of GSH and the hydroxyl oxygen of Tyr7 remained below 4.5 Å, suggesting frequent and stable interaction. GSH was stabilized in the G site within 2 ns, remaining within ~0.4 nm in chain A; chain B showed a similar stabilization. Occasionally, the thiol group temporally moved away from Tyr7 between 130 and 380 ns in both chains and again between 800 and 1000 ns in chain B. Despite these fluctuations, the interactions remained predominantly stable across the simulation.

#### 3.1.3. Site Identification

The SILCS methodology was used to identify and evaluate potential binding sites in Ts25GST, incorporating residue flexibility and solubility-based site accessibility. Analysis of the Ts25GST-GSH complex in a water/isopropanol (0.2 M) co-solvent system using VMD produced FragMaps with high-density isopropanol regions on the protein surface (Figure 3). These density maps identified high-density regions in the access groove to the catalytic site, filtered to values exceeding 0.02 ± 0.001 atoms/Å^3^ (bulk solvent density).

Free energy values were computed for each high-density region by substituting central cell occupancy values into Equation (7), using bulk solvent occupancy as the baseline (*N*_0_). Subsequently, critical points were designated where Δ*G_grid_* < 0.5 kcal/mol. A site was defined by the presence of at least three probe molecules.

Equation (7), free energy grid
(7)$$∆G_{grid}=-RTln\left(\frac{N_{i}}{N_{0}}\right)$$
where

Δ*G_grid_*: free energy of interaction in the central hotspot cell calculated using the inverse Boltzmann relation;*N_i_*: occupancy density of probe atoms in the central cell of each identified hotspot;*N*_0_: reference occupancy density calculated from probe simulations (isopropanol) in a co-solvent system with water and in the absence of protein;*R*: ideal gas constant;*T*: simulation temperature in Kelvin (K).

Three potential sites were identified, with theoretical dissociation constants (*K_D_*) calculated with Equation (8) and the percentage of identical amino acids relative to human mu-class GSTs (Conservation) shown in Table 1. Site 1, located in the groove leading to the catalytic site, showed the lowest *K_D_* (12.1 ± 6.6 mM) with a sequence identity of less than 60%. Probes interacted with residues Tyr116, Thr112, Asn209, and Gly210 while avoiding G-site residues due to the presence of GSH.

Equation (8), theoretical *K_D_*
(8)$$K_{D}=\frac{e^{\sum_{i}^{x}{∆G}_{grid}}}{RT}$$
where

*K_D_*: theoretical dissociation constant of the probe at the identified site.

### 3.2. Discovering Process

#### 3.2.1. Virtual Screening

To simulate receptor flexibility, we performed ensemble docking using conformations derived from homology models and MD simulations [25,26,27]. Five Ts25GST-GSH complex trajectories representing chain A and SILCS-identified sites were selected. Figure 4 outlines the three-stage filtering process, comparing scoring metrics from Vina and CHEMPLP. This approach identified 28 compounds with Vina binding energies ranging from −10.0 to −11.5 kcal/mol and CHEMPLP scores between 75 and 80. These compounds exhibited lower affinity for human mu-class GSTs with free energies between −7.0 and −8.5 kcal/mol and scores of 50 to 62. Re-docking to human GSTs resulted in only 10% of ligand scores being below −9.0 kcal/mol and GOLD scores of 70, aligning with a typical virtual screening success rate of 5–15%.

#### 3.2.2. Production and Kinetic Parameters of Recombinant Ts25GST

An active recombinant form of Ts25GST was successfully purified. SDS-PAGE analysis revealed a prominent band at approximately 25.0 kDa in the 20.0 mM GSH elution lane, with 12.0 µg of total protein per fraction, except the wash lane, which contained 3.0 µg due to a lower protein concentration (Figure 5A). Kinetic studies using CDNB as a substrate (Figure 5B) displayed sigmoidal behavior, in agreement with the findings of Miranda-Blancas et al. [12] for sigma-class GSTs in the same organism. Fitting these data to the Hill equation suggested positive cooperativity, with a Hill coefficient, n > 1.0, suggesting that CDNB binding at one site enhances affinity at an additional binding site.

For GSH, the kinetic behavior followed a hyperbolic Michaelis–Menten model, showing similar *V_max_* values for both substrates but a lower *K_m_* for GSH, indicating higher affinity. Positive cooperativity, commonly observed in multimeric or allosteric proteins, was further supported by a Hill coefficient near 2. This value suggests the presence of two potential CDNB binding sites per enzyme dimer. This cooperative binding implies that CDNB binding at one subunit could enhance binding at the other subunit, indicating an allosteric regulation mechanism potentially involving a non-catalytic CDNB binding site. Table 2 shows the kinetic parameters determined for Ts25GST.

#### 3.2.3. In Vitro Activity Assays

The enzyme activity was evaluated in the presence of 5.0 mM GSH and 8.0 mM CDNB, under optimal conditions established from previous kinetic studies. The screening of the 28 compounds (Figure 6A) revealed that 7 demonstrated enhanced enzymatic activity, 15 had minimal effects, and 6 inhibited activity by 20–70%. Compounds 11 and 15 showed the highest levels of inhibition, with reductions of 57.8 ± 3.3% and 68.3 ± 3.4%, respectively. When tested against HGSTM1, these two compounds showed lower inhibition percentages equivalent to 33.4 ± 2.2 and 30.8 ± 1.9 (Figure 6B), indicating greater specificity towards parasitic GST.

Structural alignment of Ts25GST with five human mu-class GST isoforms revealed over 90% sequence identity, particularly at the screening site. This high degree of conservation suggests that inhibition effects observed for HGSTM1 could potentially extend to other HGSTM isoforms if this site acts as the primary interaction target.

### 3.3. Inhibition Characterization

#### 3.3.1. IC_50_ and Inhibitor Effects

The half-maximal inhibitory concentrations (IC_50_) for selective inhibitors *i11* and *i15* were determined from the dose–response curves (Figure 7) and are summarized in Table 3. Inhibitor *i15* had a lower IC_50_ (25.7 ± 1.1 µM) than *i11* (28.8 ± 1.2 µM), indicating stronger inhibition. The two-dimensional interaction diagrams (Figure 7B,C) illustrate the binding of both inhibitors to residues in the H site, with *i11* primarily interacting with Thr112 and Tyr116, while *i15* interacts with all four residues, suggesting a potentially competitive mechanism against xenobiotics. Additional kinetic assays with varying inhibitor concentrations were conducted to refine these insights.

Kinetic data with 5.0 mM GSH and different CDNB concentrations (Figure 7D,E; Table 4 and Table 5) revealed distinct inhibition mechanisms. Inhibitor *i15* reduced *V_max_* without significantly affecting *K_m_*, while maintaining a Hill coefficient close to 2, indicating a non-competitive inhibition relative to CDNB. Conversely, *i11* increased *K_m_* without affecting *V_max_* at lower concentrations, suggesting competitive inhibition with CDNB, though results became inconsistent at 30 µM concentration. The competitive inhibition of an enzyme involves a non-substrate molecule blocking the active site. However, this approach is less desirable for pharmacological purposes, as the inhibitor’s effectiveness depends on the substrate concentration. Under physiological conditions, substrate concentrations are often high enough to reduce the inhibitor’s effectiveness.

Interestingly, the docking poses did not fully match the observed kinetic behaviors. Although both inhibitors involve site H residues, only *i11* demonstrated competitive inhibition with CDNB. Molecular dynamics (MD) simulations were conducted with complexes containing two molecules of each inhibitor to explore these discrepancies.

#### 3.3.2. Interaction Modeling

Figure 8A shows the three lowest-energy conformations for the ternary complex Ts25GST−GSH−*i11*, where the inhibitor alternates between extended and folded conformations within the active site cleft. Interaction diagrams (Appendix A) reveal that all four residues of the H site are in proximity, with *i11* interacting in three structures across the H site region, and with mu loop residues (Val35, Phe41, and Arg 43). In contrast, the corresponding lowest-energy conformations from the simulations of the ternary complex with *i15* (Figure 8B), reveal distinct ligand dynamics between chains, with *i15* in chain A showing more mobility as it adopts various conformations, while in chain B it remained in an extended conformation. Only in minimal-energy conformations did *i15* interact with the mu loop residues (Val35 and Arg43).

Differences in mu loop flexibility and GSH displacement between *i11* and *i15* were also observed. Notably, *i11* decreased the mu loop RMSF by approximately 10% compared to the Ts25GST-GSH complex, while *i15* increased RMSF by 12%, suggesting that *i11* stabilizes the mu loop. In the presence of *i11*, GSH remained stable in the G site; however, *i15* interaction displaced GSH by engaging residues like Trp8, Asp9, and Leu13.

## 4. Discussion

The AlphaFold2 model yielded robust validation metrics, with an ERRAT score exceeding 80%, indicating reliable non-bonded interactions [39], and with a Verify3D rating of 91.2%, which confirms its high residue compatibility with the expected physicochemical properties [40]. Additionally, the Ramachandran plots showed no residues with torsion angles in the disallowed regions [42]. The conservation analysis of the G site revealed that GSH maintained its typical orientation in GST-GSH complex structures, positioning the thiol group toward the activating tyrosine [13,14], a feature faithfully reproduced in our model. However, the active site appeared more open than anticipated, likely due to the template used (PDB ID: 1GSU), which was co-crystallized with S-hexylglutathione—a larger conjugate than GS-DNB—resulting in an expanded active site. To refine the structure and simulate the conformational variability of the target protein for the subsequent rigid docking of potential inhibitors, we conducted three MD simulations of the Ts25GST-GSH complex in explicit solvent. Using PCA, followed by structural clustering [43], we identified five representative conformations (Figure 2) for virtual screening. During the simulations, glutathione maintained an extended conformation, consistently orienting its thiol group toward the hydroxyl group of Tyr7, with intergroup distances conducive to stable interactions, a hallmark of GST-GSH complexes [9,10,11,12,13,14].

Using the SILCS method we identified potential small-molecule binding sites on the Ts25GST surface. The site with the highest affinity for the isopropanol probe shares 58.3% identity with the analogous residues in HGSTM1. This site is in a cleft near the catalytic region and is composed of eight hydrophobic residues, five uncharged polar residues, four positively charged residues, and three negatively charged residues. Its proximity to the catalytic region suggests there is potential for it to enhance inhibition, although it may also introduce competition with the H site, as probe interactions were observed with three H site residues.

The discovery process led to the identification of two selective inhibitors for Ts25GST, designated as *i11* and *i15*. Among them, *i15* showed superior performance in all assays, exhibiting over 10% greater in vitro inhibition compared to *i11*, as well as a lower IC50 (25.7 µM for *i15* versus 28.8 µM for *i11*), indicating a stronger binding affinity. Kinetic assays revealed that *i11* exhibited competitive inhibition with CDNB, suggesting it targets the same binding site or closely interacts with the active site residues associated with CDNB binding. In contrast, *i15* displayed non-competitive inhibition, binding at an alternate site without directly competing with CDNB. MD simulations supported these findings; *i11* was observed partially occupying the H site, forming stable interactions with several residues in the mu loop, a behavior consistent with competitive binding. Conversely, *i15* only occasionally interacted with residues at the edge of the H site and displaced GSH from the G site, corroborating its non-competitive inhibition against CDNB.

## 5. Conclusions

This study successfully identified two selective inhibitors, *i11* and *i15*, for the mu-class Ts25GST through an integrated approach combining computational and in vitro methods. These inhibitors demonstrated significant inhibitory activity, with percentages of 57.8 ± 3.3 for *i11* and 68.3 ± 3.4 for *i15* at a concentration of 40 µM. In contrast, their activity against HGSTM1 was significantly reduced, with inhibition percentages of 33.4 ± 2.2 for *i11* and 30.8 ± 1.9 for *i15*. The IC50 values were comparable, with *i11* at 28.84 ± 1.17 µM and *i15* at 25.70 ± 1.1 µM. The kinetic analysis revealed that *i11* behaves as a competitive inhibitor with CDNB, while *i15* exhibited non-competitive inhibition.

MD simulations suggested that *i11* stabilizes the mu loop, an essential region for xenobiotic binding, while *i15* displaces GSH from the G site, influencing binding interactions. The ChemBridge DiverSet library, along with other chemical diversity libraries used for hit discovery, is designed to cover a broad chemical space. This implies the existence of two distinct families of compounds related to *i11* and *i15*, within which more potent and selective analogues against Ts25GST can be found, while trying to improve their chemical stability and permeability properties to reach their target in the parasite. These findings lay the groundwork for the further optimization of selective inhibitors targeting Ts25GST, with potential therapeutic implications.

## Figures and Tables

**Figure 1 biomolecules-15-00007-f001:**
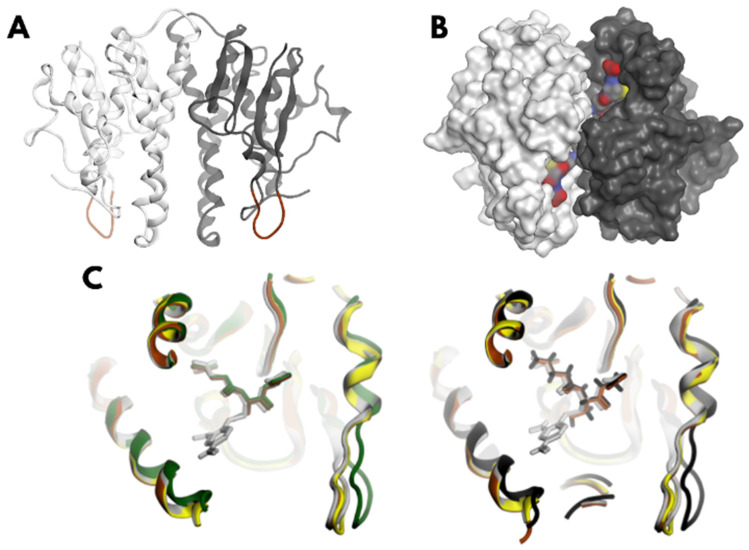
Homology model of Ts25GST built using AlphaFold2 and template PDB-ID 1GSU. (**A**) Front view of the Ts25GST model, highlighting the mu-loops in brown. Chain A is shown in light gray, and Chain B in gray. (**B**) Molecular surface representation of Ts25GST, illustrating the GSH molecules bound at the G sites. (**C**) Comparative analysis of the mu-loop from Ts25GST chain A (green) and chain B (black) with three human class Mu M1 GST structures as follows: without GSH (yellow, PDB-ID 1GTU); with GSH bound (brown, PDB-ID 1XW6); and with a GSH-CDNB conjugate (light gray, PDB-ID 1XWK).

**Figure 2 biomolecules-15-00007-f002:**
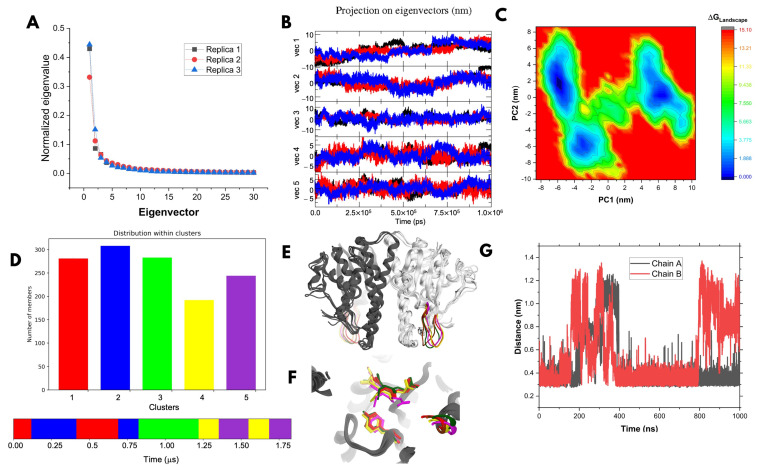
Molecular dynamics results of Ts25GST-GSH. (**A**) Normalized PCA; (**B**) 1D projection of the first five components; (**C**) free energy landscape projection on the first two components; (**D**) clustering on energy minima; (**E**) superposition of the five representative cluster structures; (**F**) close-up of the G site in the superposition of the 5 conformers; and (**G**) distance variation between the oxygen of the OH group in Tyr7 and the sulfur of the thiol in GSH.

**Figure 3 biomolecules-15-00007-f003:**
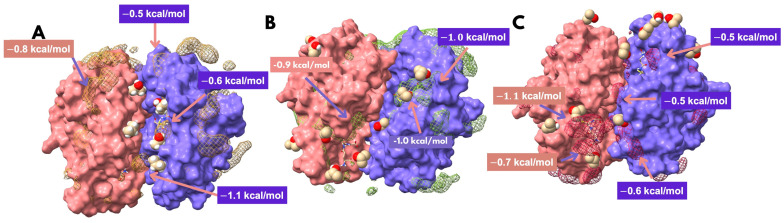
SILCS results from the three replicas (**A**–**C**) performed with the solvated model in a mixed solvent of 0.2 M isopropanol/H_2_O. The regions with the highest occupation density of the probe and the estimation of the free energy grid of the points classified as critical according to our methodology are shown.

**Figure 4 biomolecules-15-00007-f004:**
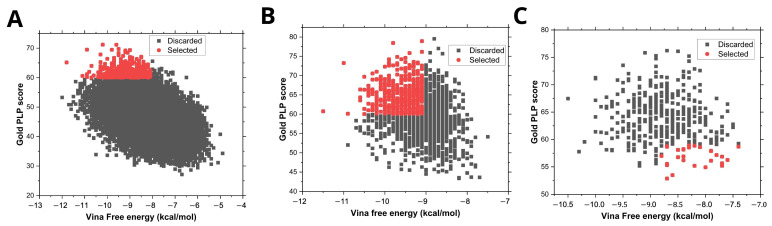
Virtual screening process. (**A**) Relaxed screening of the complete compound library; (**B**) exhaustive screening of the top results; and (**C**) screening on human mu-class GST structures (HGSTM).

**Figure 5 biomolecules-15-00007-f005:**
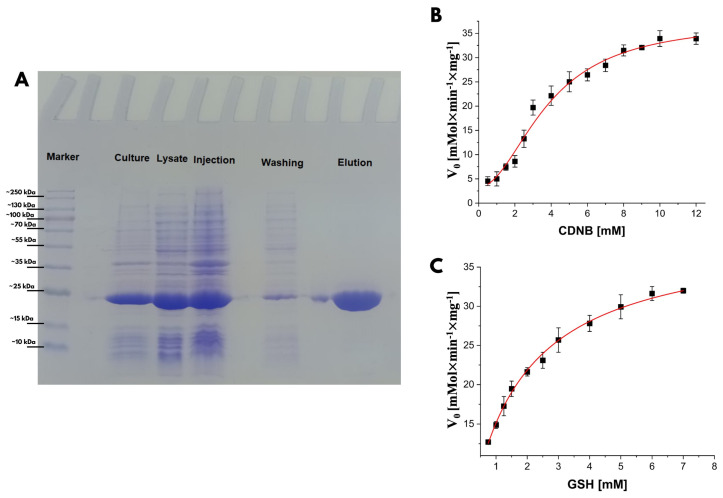
Results of the purification and enzymatic activity of recombinant Ts25GST. (**A**) PAGE-SDS of the recombinant protein. (**B**) Enzyme kinetics with variable CDNB. (**C**) Enzyme kinetics with variable GSH.

**Figure 6 biomolecules-15-00007-f006:**
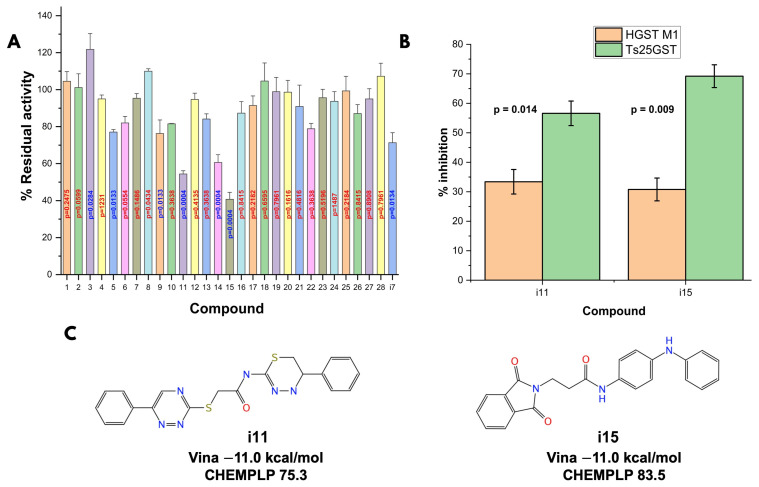
Results of the in vitro inhibitory activity assays (**A**); significance values adjusted by FDR-corrected *T*-tests [47], selectivity (**B**); and comparison of the 2D structure and docking results of the two identified inhibitors (**C**).

**Figure 7 biomolecules-15-00007-f007:**
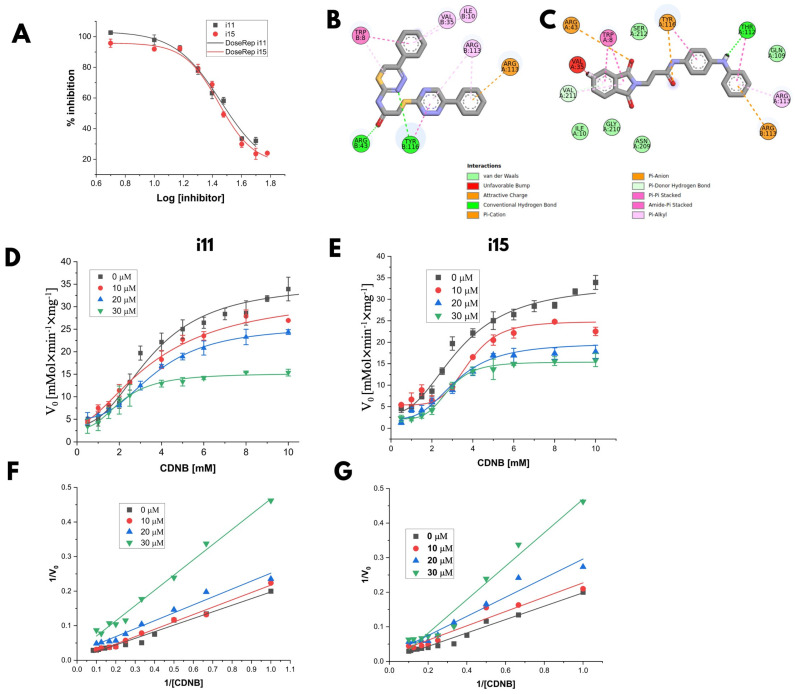
Experimental results of inhibitor characterization. (**A**) Dose–response curve for IC_50_ calculation. (**B**,**C**) Two-dimensional diagrams of the interaction between *i11* and *i15* with chain A of the Ts25GST model, respectively. Effect of the concentration of inhibitors *i11* (**D**) and *i15* (**E**) on the kinetic parameters of recombinant Ts25GST for CDNB variation. Double reciprocal plots for the reaction rates with different concentrations of CDNB and in the presence of different concentrations of inhibitors *i11* (**F**) and *i15* (**G**).

**Figure 8 biomolecules-15-00007-f008:**
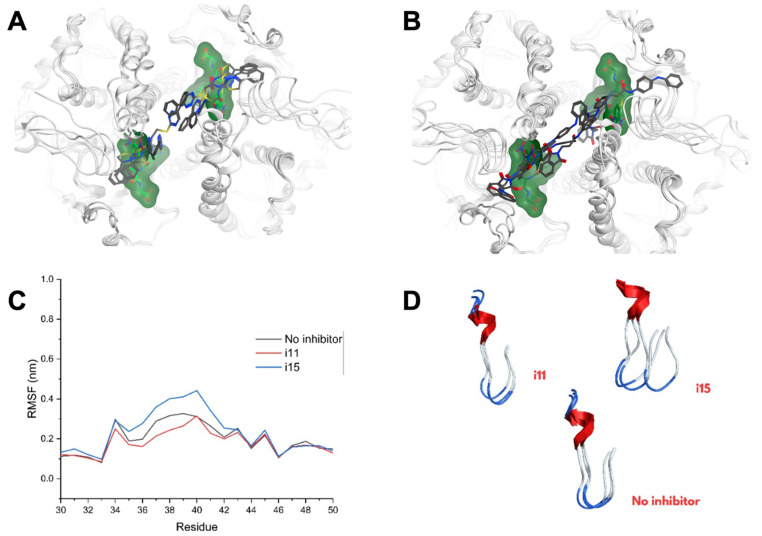
Superposition of conformations of the energy minima from the Ts25GST-GSH simulations in complex with 2 molecules of *i11* (**A**) and *i15* (**B**). Results of RMSF calculations (**C**). Analysis of the effect of the inhibitors on the mobility of the mu loop (**D**).

**Table 1 biomolecules-15-00007-t001:** Results of the theoretical *K_D_* calculation with SILCS and conservation relative to human mu-class GST structures for each site.

Site	*K_D_* (µM^−1^)	Conservation (%)
1	12.1 ± 6.6	58.3%
2	2127 ± 1252	60.0%
3	56.3 ± 21.9	47.5%

**Table 2 biomolecules-15-00007-t002:** Kinetic parameters of recombinant Ts25GST.

Kinetic Parameter	CDNB	GSH
*K_M_*	3.5 ± 0.3 mM^−1^	1.60 ± 0.1 mM^−1^
*V_max_*	39.4 ± 2.0	39.3 ± 0.3
*n*	2.4 ± 0.3	-
R^2^	0.99	0.99
Fitting model	Hill	Michaelis–Menten

**Table 3 biomolecules-15-00007-t003:** IC50 calculation for *i11* and *i15*.

Compound	IC_50_ [µM]	R^2^
*i11*	28.8 ± 1.2	0.98
*i15*	25.7 ± 1.1	0.99

**Table 4 biomolecules-15-00007-t004:** Effect of *i11* concentration on the kinetic parameters of recombinant Ts25GST.

Kinetic Parameter	0 µM	10 µM	20 µM	30 µM
*K_M_*	3.5 ± 0.1	4.5 ± 0.7	6.4 ± 0.2	2.4 ± 0.2
*V_max_*	39.4 ± 2.0	40.7 ± 0.4	40.6 ± 0.4	17.5 ± 1.3
*n*	2.3 ± 0.3	1.8 ± 0.4	2.7 ± 0.2	2.6 ± 0.3
R^2^	0.99	0.98	0.98	0.96

**Table 5 biomolecules-15-00007-t005:** Effect of *i15* concentration on the kinetic parameters of recombinant Ts25GST.

Kinetic Parameter	0 µM	10 µM	20 µM	30 µM
*K_M_*	3.5 ± 0.3	3.4 + 0.5	3.3 + 0.4	3.3 + 0.7
*V_max_*	39.4 ± 2.0	32.9 + 1.8	25.8 + 2.3	15.3 + 2.1
*n*	2.3 ± 0.3	1.8 + 0.4	1.7 + 0.3	2.8 + 0.6
R^2	0.99	0.99	0.99	0.98

## Data Availability

The original contributions presented in this study are included in the Appendix A. Further inquiries can be directed to the corresponding authors.

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
