# Peer review of "Discovery and Characterization of Two Selective Inhibitors for a Mu-Class Glutathione S-Transferase of 25 kDa from Taenia solium Using Computational and Bioinformatics Tools"

_biomolecules, 2024, doi:10.3390/biom15010007_

Round 1
Reviewer 1 Report
Comments and Suggestions for Authors
The manuscript "Discovery and characterization of two selective inhibitors for a mu-class glutathione transferase of 25 kDa from Taenia solium using computational and bioinformatics tools" reports the identification of two inhibitors based on docking of a compound library and experimental verification of the inhibition in an in vitro assay. The selectivity was evaluated by comparing the inhibition of Ts25GST with the inhibition of human mu-class GSTM1.
The story is sound and the compounds seem to be valuable to be further analyzed/optimized. Overall the manuscript is well written. I however see some issues and some polishing is required before the manuscript can be considered for publication.
Please have a detailed look at all the figure captions. There are likely some mistakes that need revision:
Figure 1B does not show an identity matrix as the text says, the description for 1C also seems to be wrong. Please check the rest of the caption as well. In figure 2, 2F is not correctly described and 2G is not described below the figure. In the figure 3 caption A, B and C are not mentioned. In the caption of figure 5 the description of 5B (variation of CDNB) and 5C may need to be interchanged. In figure 6 the "B" and the "D" are not visible in the figure. Did you use i11 in figure 7D and i15 in 7E?
In figure 6A it looks like compounds I10 and I14 have the highest inhibition. Is that correct, did I misunderstand the labels?
Please be more specific about the identity of the assay conditions for human mu-class GSTM1 and the quality of the purchased GSTM1. You state that "Selectivity tests were performed similarly". The assay conditions and sample characteristics are essential to allow a solid and sound comparison with Ts25GST.
Additional questions and comments:
Introduction: Please explain in a bit more detail which reactions are catalyzed by gluthathion-S-transferases and how the active site looks like.
Can reduced activity of one GST be rather well compensated by other GSTs (in T. solium) or are they essential? Maybe the discussion can be expanded in this direction.
Which spectrophotometer is used for the enzyme kinetics investigation?
Figure 5A: Please add molecular weights to the marker or specify which marker was used to allow a molecular weight estimate based on the figure.
Figure 8 looks a bit crowded with some details in 8C and D that are nearly too small to be well readable.
The list of textbook equations in a separate results section 3.4 comes out of nowhere. Please consider to move the equations to the respective part of the methods section or the supplement.
Is it possible to provide Ki values for both inhibitors based on the data you have? IC50 has the disadvantage that it is relative to the individual concentrations.
Author Response
REVIEWER 1
The manuscript "Discovery and characterization of two selective inhibitors for a mu-class glutathione transferase of 25 kDa from Taenia solium using computational and bioinformatics tools" reports the identification of two inhibitors based on docking of a compound library and experimental verification of the inhibition in an in vitro assay. The selectivity was evaluated by comparing the inhibition of Ts25GST with the inhibition of human mu-class GSTM1.
The story is sound and the compounds seem to be valuable to be further analyzed/optimized. Overall the manuscript is well written. I however see some issues and some polishing is required before the manuscript can be considered for publication.
Please have a detailed look at all the figure captions. There are likely some mistakes that need revision:
1.- Figure 1B does not show an identity matrix as the text says, the description for 1C also seems to be wrong. Please check the rest of the caption as well. In figure 2, 2F is not correctly described and 2G is not described below the figure. In the figure 3 caption A, B and C are not mentioned. In the caption of figure 5 the description of 5B (variation of CDNB) and 5C may need to be interchanged. In figure 6 the "B" and the "D" are not visible in the figure. Did you use i11 in figure 7D and i15 in 7E?
In the new version of the manuscript, figures 1, 2, 6 and 8 were modified and all the errors mentioned were corrected.
2.- In figure 6A it looks like compounds I10 and I14 have the highest inhibition. Is that correct, did I misunderstand the labels?
A missing compound shifted the order of the inhibitors in the figure. The new figure 6A corrects this, and significance levels for each set of activity measurements are now included.
3.- Please be more specific about the identity of the assay conditions for human mu-class GSTM1 and the quality of the purchased GSTM1. You state that "Selectivity tests were performed similarly". The assay conditions and sample characteristics are essential to allow a solid and sound comparison with Ts25GST.
It was now specified in section 2.7, lines 176-178, that human GST M1 was obtained from Oxford Biomedical Research and that the assays were performed under conditions identical to those used with Ts25GST.
Additional questions and comments:
4.- Introduction: Please explain in a bit more detail which reactions are catalyzed by gluthathion-S-transferases and how the active site looks like.
This is now described on the second page of the manuscript, second paragraph, lines 54 to 61.
5.- Can reduced activity of one GST be rather well compensated by other GSTs (in T. solium) or are they essential? Maybe the discussion can be expanded in this direction.
Although the concentration of the different classes of cytosolic GST is different, the nature and dimension of their H site is variable and characteristic of each class. For example, it has been observed that Ts25GST actively participates in the reduction of membrane lipid oxidation products. Besides, we mentioned that each class of cGST of T. solium participates in other essential functions for the parasite. This has led us to search for inhibitors for each of them.
6.- Which spectrophotometer is used for the enzyme kinetics investigation?
We used a double beam spectrophotometer Shimadzu UV-1800. This was mentioned in lines 144-145.
7.- Figure 5A: Please add molecular weights to the marker or specify which marker was used to allow a molecular weight estimate based on the figure.
Molecular masses of markers were added to the figure.
8.- Figure 8 looks a bit crowded with some details in 8C and D that are nearly too small to be well readable.
This figure was reworked, changing some colors and transferring Figures 8C and 8D (2D interaction diagrams) to Supplementary Materials as Figure S3.
9.- The list of textbook equations in a separate results section 3.4 comes out of nowhere. Please consider to move the equations to the respective part of the methods section or the supplement.
The equations have been moved to the respective sections where they are mentioned.
10.- Is it possible to provide Ki values for both inhibitors based on the data you have? IC50 has the disadvantage that it is relative to the individual concentrations.
Yes, we can calculate the Ki values for both inhibitors with respect to the CDNB variation, however we do not yet have the data to calculate the Ki with respect to GSH and we believe that it is necessary to have both values in order to give a more reliable interpretation.
Reviewer 2 Report
Comments and Suggestions for Authors
- The bioinformatic approach appears to be thorough and satisfactory, but as stated in the instructions to authors (“experimental data”, https://www.mdpi.com/journal/biomolecules/instructions#preparation), “all compounds should be mentioned by correct chemical name, followed by any numerals used to refer to them in the paper”. For example, the similar previous study by García-Gutiérrez et al that is cited has a supplementary table with all structure formulas, names, IDs and structures, which is essential for any readers to perform follow-up work and to evaluate the results. For example, did the 28 compounds and the shorter list of active compounds share any chemical structures that would be useful for SAR? Based on that, can analogs be further optimized for selectivity of the parasite vs the host? How similar are i11 and i15 in structure to the previously identified i7? None of this can be ascertained from the current manuscript.
- The “Data Availability Statement” provided is just the instructions to the authors. As described above, the data availability section should include supplementary materials with full chemical structures, docking scores, and raw experimental data to enhance transparency and reproducibility.
- Given the similarity in approach to the García-Gutiérrez et al. study, but targeting a different taenia GST, it would have been very useful for the authors to also perform in vitro screening using their previously-identified i7 compound that was effective against Ts26GST. It is not clear at all whether the previous i7 or the two new prioritized compounds have specificity against their specific TsGSTs or whether they may both target all TsGSTs. With the high level of activity against HGSTM1, it seems likely that it also may target the other TsGSTs.
- Also, since the study is similar to the previous García-Gutiérrez one, in the discussion and conclusion, the authors should more directly state how their approach refines or builds upon the results form that study, including highlighting the biological or therapeutic significance of studying Ts25GST relative to TsGST26.
- Data in Figure 6A and 6B lack a statistical analysis. For 6A, FDR-corrected T-tests could be used to compare each compound to control, and 6B could just be a HGST vs Ts25GST comparison.
- I am confused about what is being shown in Figure 6A. Compound i11 is shown to have near-zero % inhibition, and i13 is less than 60%. The numbers stated in the text correspond to 6B, but why was 11 selected based on the results from 6A when it is almost zero? Was it based on other results, and why is it so much higher in 6B?
- Based on the data in 6A, the authors should consider clustering the 28 compounds based on their chemical scaffolds and performing a SAR analysis to identify structural trends linked to activity. This could provide a basis for rational design of more potent and selective analogs.
- While it’s recognized that the human GST has lower % inhibition values, they are still quite high at 33% and 30% respectively, potentially making them quite cytotoxic and dangerous for use. A human cell line cytotoxicity assay and ideally an in vitro taenia motility assay would help to assure that there is good specificity against the worms.
- In vivo, there will be additional concerns about compound permeability into the worms, potential rapid degradation once in the worm, and the possibility that inhibiting the target may not be lethal against the worm anyway. While these are not expected to all be addressed in the first paper to identify compounds, they should be at least discussed in the discussion/conclusion.
Author Response
REVIEWER 2
1.- The bioinformatic approach appears to be thorough and satisfactory, but as stated in the instructions to authors (“experimental data”, https://www.mdpi.com/journal/biomolecules/instructions#preparation), “all compounds should be mentioned by correct chemical name, followed by any numerals used to refer to them in the paper”. For example, the similar previous study by García-Gutiérrez et al that is cited has a supplementary table with all structure formulas, names, IDs and structures, which is essential for any readers to perform follow-up work and to evaluate the results. For example, did the 28 compounds and the shorter list of active compounds share any chemical structures that would be useful for SAR? Based on that, can analogs be further optimized for selectivity of the parasite vs the host? How similar are i11 and i15 in structure to the previously identified i7? None of this can be ascertained from the current manuscript.
We have included in the Supplementary Material the Table S3 listing the 28 compounds tested, including their library numbers, commercial I.D., Smiles codes, Vina and Gold scores for Ts25GST and HGSTM1, log P, molecular weights, and separately their two-dimensional structures in Figure S4.
2.- The “Data Availability Statement” provided is just the instructions to the authors. As described above, the data availability section should include supplementary materials with full chemical structures, docking scores, and raw experimental data to enhance transparency and reproducibility.
We have now included Supplementary Material such as the list of tested compounds, GST activity data with varying GSH and CDNB, 2D diagrams of interactions of Ts25GST with i11 and i15.
3.- Given the similarity in approach to the García-Gutiérrez et al. study, but targeting a different taenia GST, it would have been very useful for the authors to also perform in vitro screening using their previously-identified i7 compound that was effective against Ts26GST. It is not clear at all whether the previous i7 or the two new prioritized compounds have specificity against their specific TsGSTs or whether they may both target all TsGSTs. With the high level of activity against HGSTM1, it seems likely that it also may target the other TsGSTs.
Inhibition assays of i7 against Ts25GST were performed, observing a residual activity of 70%, which is reported in Figure 6A.
4.- Also, since the study is similar to the previous García-Gutiérrez one, in the discussion and conclusion, the authors should more directly state how their approach refines or builds upon the results form that study, including highlighting the biological or therapeutic significance of studying Ts25GST relative to TsGST26.
We are interested in discovering specific inhibitors for each of the GSTs in the cytosol of T. solium, in order to subsequently try to find a compound that is capable of inhibiting all of them.
5.- Data in Figure 6A and 6B lack a statistical analysis. For 6A, FDR-corrected T-tests could be used to compare each compound to control, and 6B could just be a HGST vs Ts25GST comparison.
We appreciate the suggestion, which we followed. The significance levels for each set of activity measurements using FDR-corrected tests are now presented in Figure 6A and 6B.
6.- I am confused about what is being shown in Figure 6A. Compound i11 is shown to have near-zero % inhibition, and i13 is less than 60%. The numbers stated in the text correspond to 6B, but why was 11 selected based on the results from 6A when it is almost zero? Was it based on other results, and why is it so much higher in 6B?
In Figure 6A, a missing compound changed the order of the inhibitors. This has been corrected and the data are now presented as % residual activity with significance levels shown for each set of activity measurements.
7.- Based on the data in 6A, the authors should consider clustering the 28 compounds based on their chemical scaffolds and performing a SAR analysis to identify structural trends linked to activity. This could provide a basis for rational design of more potent and selective analogs.
We are conducting related studies, also involving i11 and i15 analogues to identify functional groups that increase affinity, but we consider this to be another work in the optimization stage.
8.- While it’s recognized that the human GST has lower % inhibition values, they are still quite high at 33% and 30% respectively, potentially making them quite cytotoxic and dangerous for use. A human cell line cytotoxicity assay and ideally an in vitro taenia motility assay would help to assure that there is good specificity against the worms.
Although the inhibitors do show an effect on human GST, we are at an early stage in the development of these compounds. Our goal is to improve the affinity for Ts25GST and, at the same time, the specificity in relation to human GSTs. We have ongoing experiments to achieve this.
9.- In vivo, there will be additional concerns about compound permeability into the worms, potential rapid degradation once in the worm, and the possibility that inhibiting the target may not be lethal against the worm anyway. While these are not expected to all be addressed in the first paper to identify compounds, they should be at least discussed in the discussion/conclusion.
We share your concern, however, at this stage of discovery we are looking for compounds capable of inhibiting our target with high efficiency and good specificity, to subsequently focus on improving their pharmacokinetic and pharmacodynamic properties.
Reviewer 3 Report
Comments and Suggestions for Authors
Current manuscript highlight the role of two novel inhibitors (i11 and i15) against Ts25GST enzyme. The enzyme is involved in detoxifying xenobiotics and managing oxidative stress in helminths during infection. Although the content related to this manuscript is good but the paper is very poorly written and there are thousands of errors and difficult to understand each paragraph. I will suggest to re-write the manuscript and resubmit for better understanding.
For instance I just highlighted a few in early part of paper but after that it was very difficult to follow the direction because of poor flow...
Abstract:
1. GST is Glutathione S-transferase rather than Glutathione transferase.
Introduction:
2. Line 53: It is not clear from the sentence that GSH and CDNB both are substrates of GST and i7 inhibits the binding.
3. Introduction is poorly written. Authors should highlight the basic mechanism of role of glutathione as well as CDNB in parasite body. Authors mentioned CDNB is a substrate but it is not clear what does it do and what is the source of this substrate with respect to parasitic infection.
Material and method:
4. In homology Modeling section, it is not clear why authors used three structure rather than one to put GSH at the binding site. Also what are the criteria to generate 5 independent structures from alphafold-2.
5. Authors mentioned they used 20% DMSO for enzymatic assay. The amount of organic solvent DMSO is much higher and it may affect the denaturation of proteins. So I would suggest to do the assay at lower DMSO concentration or any evidence that protein does not alter its structure in presence of this high quantity of DMSO.
6. Why two different fitting models for GSH and CDNB? (Hill model vs Michaelis-Menten model)
7. In section 2.6 (enzyme kinetics) nothing mentioned about GST (I think this is the main enzyme whose rate need to be calculated)
Results:
8. I suggest to mention GSH-binding site and xenobiotic binding site in the structure for clear understanding. Also figure 1 is not so informative and missing multiple labellings. Color combination in comparison one is very poor and very difficult to observe the similarity or differences.
9. Resolution is very poor and very difficult to see the text.
10. Correct figure legend 1 (it is completely off in wordings)
11. Figure 2: I do not see any A2,A3….It is completely weird the text does not match with figure.
Author Response
REVIEWER 3
Current manuscript highlight the role of two novel inhibitors (i11 and i15) against Ts25GST enzyme. The enzyme is involved in detoxifying xenobiotics and managing oxidative stress in helminths during infection. Although the content related to this manuscript is good but the paper is very poorly written and there are thousands of errors and difficult to understand each paragraph. I will suggest to re-write the manuscript and resubmit for better understanding.
For instance I just highlighted a few in early part of paper but after that it was very difficult to follow the direction because of poor flow...
Abstract:
- GST is Glutathione S-transferase rather than Glutathione transferase.
In all our works we follow the nomenclature proposed by Mannervik B, et al., Nomenclature for mammalian soluble glutathione transferases. Methods Enzymol. 2005;401:1-8. doi: 10.1016/S0076-6879(05)01001-3. But, we agree to change the nomenclature to Glutathione S-transferase.
Introduction:
- Line 53: It is not clear from the sentence that GSH and CDNB both are substrates of GST and i7 inhibits the binding.
That paragraph was rewritten to now describe the GST conjugation reaction, the substrates, and the location of their binding sites on lines 54 to 61, page 2.
- Introduction is poorly written. Authors should highlight the basic mechanism of role of glutathione as well as CDNB in parasite body. Authors mentioned CDNB is a substrate but it is not clear what does it do and what is the source of this substrate with respect to parasitic infection.
We have modified the introduction to mention that CDNB is a universal substrate. This synthetic, hydrophobic and electrophilic molecule is conjugated to glutathione by all types of GSTs, cytosolic, mitochondrial and microsomal, and is therefore used in standard assays to measure transferase activity.
Material and method:
- In homology Modeling section, it is not clear why authors used three structure rather than one to put GSH at the binding site. Also what are the criteria to generate 5 independent structures from alphafold-2.
Alphafold-2 automatically generates 5 independent models and we chose the one with the best structural assessment indicators. To position GSH in our Ts25GST model, we superimposed its structure with three mu-class GST complexes bound to GSH to demonstrate the high conservation of the G site. However, for the final model of the Ts25GST-GSH complex, we chose the structure with the best resolution as a template. This was specified in the new version of the manuscript on lines 94-96.
- Authors mentioned they used 20% DMSO for enzymatic assay. The amount of organic solvent DMSO is much higher and it may affect the denaturation of proteins. So I would suggest to do the assay at lower DMSO concentration or any evidence that protein does not alter its structure in presence of this high quantity of DMSO.
The enzymatic activity of Ts25GST was measured at different concentrations of DMSO (5, 10, 20 and 30%) and it was observed that there are no significant differences with respect to the enzymatic activity in the absence of DMSO. This result is shown in the Supplementary Material (Figure S4).
- Why two different fitting models for GSH and CDNB? (Hill model vs Michaelis-Menten model)
The Michaelis-Menten model is generally applied when the activity curve shows hyperbolic behavior, however, when the pattern is sigmoid, the existence of cooperativity between the catalytic sites or the existence of allosteric sites is presumed, so the Hill equation is then used as a fitting model.
- In section 2.6 (enzyme kinetics) nothing mentioned about GST (I think this is the main enzyme whose rate need to be calculated)
We have modified the beginning of section 2.6 to indicate that the activity measured is of glutathione S-transferase.
Results:
- I suggest to mention GSH-binding site and xenobiotic binding site in the structure for clear understanding. Also figure 1 is not so informative and missing multiple labellings. Color combination in comparison one is very poor and very difficult to observe the similarity or differences.
We modified Figure 1 to more clearly indicate the location of the GSH and xenobiotic binding sites, also changing the colors to create a better contrast.
- Resolution is very poor and very difficult to see the text.
Figures with 2D interaction representations containing very small labels were improved and moved to the Supplementary Material.
- Correct figure legend 1 (it is completely off in wordings)
Figure 1 was modified and the figure captions were revised and corrected.
- Figure 2: I do not see any A2,A3….It is completely weird the text does not match with figure.
The labels referred to figures in the Appendix A. In the new version, these figures have been labeled as part of the Supplementary Material, where they can be found as Figures S2A, S2B, etc.
Round 2
Reviewer 1 Report
Comments and Suggestions for Authors
The authors did a good job in revising the manuscript. In particular the spotted mistakes were corrected and the presentation of the results was improved. I consequently recommend the manuscript for publication.
Reviewer 2 Report
Comments and Suggestions for Authors
The authors have addressed important concerns, especially in regards to data transparency, which has substantially improved the value of the manuscript for readers.